# Synthesis of a New Dinuclear Cu(I) Complex with a Triazine Ligand and Diphenylphosphine Methane: X-ray Structure, Optical Properties, DFT Calculations, and Application in DSSCs

Carlos A. Peñuelas [1], José J. Campos-Gaxiola [1,*], Rody Soto-Rojo [1], Adriana Cruz-Enríquez [1],
Edgar A. Reynoso-Soto [2], Valentín Miranda-Soto [2], Juventino J. García [3], Marcos Flores-Álamo [3],
Jesús Baldenebro-López [1] and Daniel Glossman-Mitnik [4]

[1] Facultad de Ingeniería Mochis, Universidad Autónoma de Sinaloa, Fuente de Poseidón y Prol. A. Flores S/N, C.U., Los Mochis C.P. 81223, Mexico; carlos.apg.fim@uas.edu.mx (C.A.P.); rody.soto@uas.edu.mx (R.S.-R.); cruzadriana@uas.edu.mx (A.C.-E.); jesus.baldenebro@uas.edu.mx (J.B.-L.)
[2] Centro de Graduados e Investigación en Química, Instituto Tecnológico de Tijuana, Tecnológico Nacional de México, Apartado Postal 1166, Tijuana C.P. 22000, Mexico; edgar.reynoso@tectijuana.edu.mx (E.A.R.-S.); vmiranda@tectijuana.mx (V.M.-S.)
[3] Facultad de Química, Universidad Nacional Autónoma de México, Cto. Exterior S/N, C.U., Ciudad de México C.P. 04510, Mexico; juvent@unam.mx (J.J.G.); mfa@unam.mx (M.F.-Á.)
[4] Centro de Investigación en Materiales Avanzados, S. C., Miguel de Cervantes 120, Complejo Industrial Chihuahua, Chihuahua C.P. 31136, Mexico; daniel.glossman@cimav.edu.mx
* Correspondence: gaxiolajose@uas.edu.mx

**Abstract:** A new copper(I) complex, $[Cu_2(L)_2dppm](PF_6)_2$ (**1**) [L = 3-(2-Pyridyl)-5,6-diphenyl-1,2,4-triazine and dppm: Bis(diphenylphosphino)methane], was prepared and characterized by IR, $^1$H-NMR, $^{31}$P-NMR spectroscopy, elemental and thermogravimetric analysis, and a single-crystal X-ray diffraction technique. Complex **1** is a dinuclear compound, showing that L and dppm act as tridentate and bidentate chelating ligands, respectively. The two Cu(I) atoms exhibit a distorted tetrahedral coordination sphere embedded in $N_3P$ environments. The supramolecular interactions in the solid-state structure are characterized by $C-H\cdots N$, $C-H\cdots F$, $C-H\cdots\pi$ and $\pi\cdots\pi$ intermolecular interactions, which we studied using Hirshfeld surface and fingerprint tools. Additionally, the complex was studied experimentally using UV–Vis spectroscopy and cyclic voltammetry, and theoretical studies with time-dependent density functional theory (TD-DFT) were performed. Moreover, the optical and electrochemical properties were studied, focusing on the band gap. Compound **1** was used as a co-sensitizer in a dye-sensitized solar cell, showing a good photovoltaic performance of 2.03% ($Jsc$ = 5.095 mAcm$^{-2}$, $Voc$ = 757 mV, and $FF$ = 52.7%) under 100 mW cm$^{-2}$ (AM 1.5G) solar irradiation, which is similar to that of DSSC, which was only sensitized by N719 (2.2%) under the same condition.

**Keywords:** copper (I); triazine; phosphine; crystal structure; theoretical calculations; co-sensitized

## 1. Introduction

With the acceleration of industrialization, the demand for energy in today's society is increasing. Energy production has been largely based on fossil fuels, generating large quantities of carbon dioxide and being non-renewable. This pollution globally threatens the future of the planet [1,2]. Regarding energy and environmental issues, solar energy-to-electricity transformation technologies (photovoltaics) represent the priority alternative for green energy, triggering a speedy development of research within this area [3,4]. Photovoltaic systems are a relevant option since sunlight is the most abundant renewable resource [5,6], and photovoltaic devices can easily be integrated into buildings, providing high conversion efficiencies. In photovoltaic technology, of particular relevance are dye-sensitized solar cells (DSSCs), reported by Grätzel and O'Regan in 1991 [7]. There are two

types of these devices based on a sensitizer: organic dyes (purely organic compound) [8] and inorganic dyes (organometallic or coordination complex [9,10]. The compounds in DSSCs are in charge of light capture and the transfer of electrons into the conduction band (CB) of a semiconductor device (typically $TiO_2$), by which these compounds are supported [11]. The efficiency of the solar device also depends on the redox system, the electrolyte, and the corresponding dye properties [12]. Metal complexes possess some advantages over organic photosensitizers, and they usually exhibit higher stability. In this context, ruthenium (II) complexes such as N719 and N3 deserve particular attention due to their properties and possible applications [13,14]. Much effort has been dedicated to searching for materials to improve DSSCs' overall efficiency [15,16]. Recent record efficiencies of over 11.9–20% [17,18] were documented using dye N719, which is usually used as a reference in the arena of dyes for DSSCs.

Nevertheless, using ruthenium compounds as photosensitizers might have been a disadvantage due to the low abundance of this metal on the Earth's crust (ca. 0.001 ppm) [19] and the fact it is expensive, raising concerns about the technology's sustainability and commercial viability. Consequently, several researchers directed their efforts to the search for photosensitizers based on other metal centers, which would be of lower cost [19,20]. The use of cheaper and low-toxicity metals, namely copper or zinc, as replacements for the abovementioned expensive ruthenium(II) complexes has incentivized research in this area [10].

Since copper is an abundant element (ca. 50 ppm) [19], copper(I) centers possess a $d^{10}$ electron configuration and a favored tetracoordinate geometry [21]. Complexes with two ligands containing 2,2'-bipyridine or 1,10-phenanthroline metal-binding domains include similar photophysical properties to those of ruthenium(II) sensitizers. These properties triggered the use of Cu(I) compounds as suitable materials for DSSCs, e.g., as hole-transporting materials (HTM) [22], additives [23], and dyes [24].

Sauvage synthesized a series of homoleptic copper(I) complexes [25] of the type $[Cu(N\hat{}N)_2]^+$ with bpy ligands containing carboxylic acids as anchoring groups as well as dyes containing big band-gap semiconductors ($TiO_2$ and ZnO) useful for DSSCs. He reported a PCE that corresponds to 23.7%, relative to a device regarding ruthenium(II) dye N719. Since, relevant improvements has been made in the development of homoleptic compounds of the type $[Cu(N\hat{}N)_2]^+$ and heteroleptic $[Cu(N\hat{}N)(N\hat{}N)']^+$ or $[Cu(N\hat{}N)(P\hat{}P)]^+$ sensitizers (N̂N = diimine chelating ligand; P̂P = diphosphines chelating ligand) in dye-sensitized solar cells [26–28].

Our research groups reported previous theoretical and experimental studies of the photophysical and electrochemical properties of Cu(I) compounds containing tri-phenylphosphine ($PPh_3$) as a bulky P-donor ligand along with cis-(±)-2,4,5-tris(2-pyridyl)imidazoline or 2,4,6-tris(2-pyridyl)triazine or pyridine-2,5-dicarboxylic acid as the anchoring ligand. We documented their performance as co-sensitizers in DSSCs, achieving an FF ranging from 27.9% to 57.9%, an efficiency of 0.50–2.92%, and an $\eta_{rel}$ to N719 of 30.5–63.6% [29,30]. Optical and electrochemical studies suggest that co-adsorbents can be employed to capture light in the low-wavelength region of the visible region, overcoming the competition of light absorption by $I^-/I_3^-$, reducing charge recombination, and increasing electron lifetime. Furthermore, these materials possess moderate adsorption in the low-wavelength region (300–450 nm) and are easier to produce in good yields, reducing the price of Ru-based dyes. Therefore, they are vital materials to be used as co-sensitizers for DSSCs, assisting in the fabrication of new DSSCs with significantly low cost and a higher availability of the Earth-abundant copper-based precursors. In this paper, we document the molecular and crystal structures of one new dinuclear complex of composition $[Cu_2(L)_2dppm](PF_6)_2$ (1) [L = 3-(2-Pyridyl)-5,6 di phe-nyl-1,2,4-triazine and dppm: Bis(diphenylphosphino)methane] (see Scheme 1). The compound exhibits relevant optical and electrochemical properties, assessed in solution by UV–Vis spectroscopy and cyclic voltammetry, and further studied by quantum chemical calculations. In addition, their efficiency as co-sensitizers in DSSCs was also assessed.

**Scheme 1.** Synthetic route for complex **1**.

## 2. Results and Discussion

A combination of 3-(2-Pyridyl)-5,6-diphenyl-1,2,4-triazine (L) and Bis(diphenylphosphino)methane (dppm) with Cu(MeCN)$_4$PF$_6$ provided a dinuclear Cu(I) complex of composition [Cu$_2$(L)$_2$dppm](PF$_6$)$_2$ (**1**) (see Supplementary Material). The phosphine ligands were relevant in stabilizing the molecular structure [31]. The compound was characterized by elemental analysis; IR, $^1$H NMR, $^{31}$P{$^1$H} NMR (Figures S1–S3 in Supplementary Materials) and UV–Vis spectroscopy; thermogravimetric analysis (TGA); and single-crystal X-ray diffraction (scXRD) determinations. Additionally, the electrochemical properties of the compound were assessed with cyclovoltammetry.

### 2.1. IR and NMR Analysis

The infrared (IR) spectrum of the title compound is in good agreement with the results of the X-ray structure analyses. The spectrum exhibits characteristic C-H stretching vibrations of the aromatic rings in the range of 3055–3050 cm$^{-1}$, and the stretching vibrations of the C=N$_{imino}$ groups are C=N$_{triazine}$ 1600 cm$^{-1}$ and C=N$_{py}$ 1511 cm$^{-1}$, which are shifted to higher frequencies (~15 cm$^{-1}$) in comparison with free L due to the formation of the N→Cu bond (see Table 1 and Figure S1). The signal for the C=C stretching bands of the pyridyl and phenyl groups appear at 1481–1436 cm$^{-1}$. The band around 1436 cm$^{-1}$ is typical for the P-C$_{Ar}$ vibration of the phosphine ligand, and other bands in the 1000 and 500 cm$^{-1}$ regions are attributed to out-of-plane bending modes for the C-H, C-C, and C-N bonds. Complex **1** also exhibits a band corresponding to the asymmetric stretching vibration of the PF$_6^-$ group at 838 cm$^{-1}$ [32–34].

**Table 1.** Main signals (cm$^{-1}$) in the infrared spectra of ligand L, ppm, and complex [Cu$_2$(L)$_2$dppm](PF$_6$)$_2$ *.

| Compound | $\nu$ (C-H) | $\nu$ (C=N$_{imino}$) | $\nu$ (C=C) | $\nu$ (P-C) | $\nu$ (P-F) |
|---|---|---|---|---|---|
| L | 3050 (w) | 1579 (m) <br> 1502 (s) | 1483 (m) | – | – |
| Dppm | 3055 (w) | – | 1581 (w) <br> 1479 (m) | 1429 (m) | – |
| [Cu$_2$(L)$_2$dppm](PF$_6$)$_2$ | 3054 (w) | 1600 (m), <br> 1511 (s) | 1481 (m) | 1436 (m) | 838 (s) |

* w = weak, m = medium, s = strong.

The $^1$H NMR study of the title complex (Figure S2, Supplementary Material) displays broadened signals for both coordinated L and dppm. The spectrum shows five signals at 8.80–7.31 ppm (48 H) assigned to the L and dppm ligands. The aliphatic hydrogens in the dppm were assigned at 3.93 ppm as a triplet (2H). The $^{31}$P NMR spectrum of the compound showed a broadened signal close to −7.42 ppm (see Figure S3), which was assigned to the dppm, in addition to a septet at −144.67 ppm arising from the PF$_6^-$ anion [30,31,33].

### 2.2. X-ray Crystallography

Complex 1 was also characterized by single-crystal X-ray diffraction analysis. The molecular structure with atom labeling is represented in Figure 1. A few selected bond

lengths and angles are included in Table 2. The corresponding hydrogen bonding geometries can be found in Table S1 (see Supplementary Material).

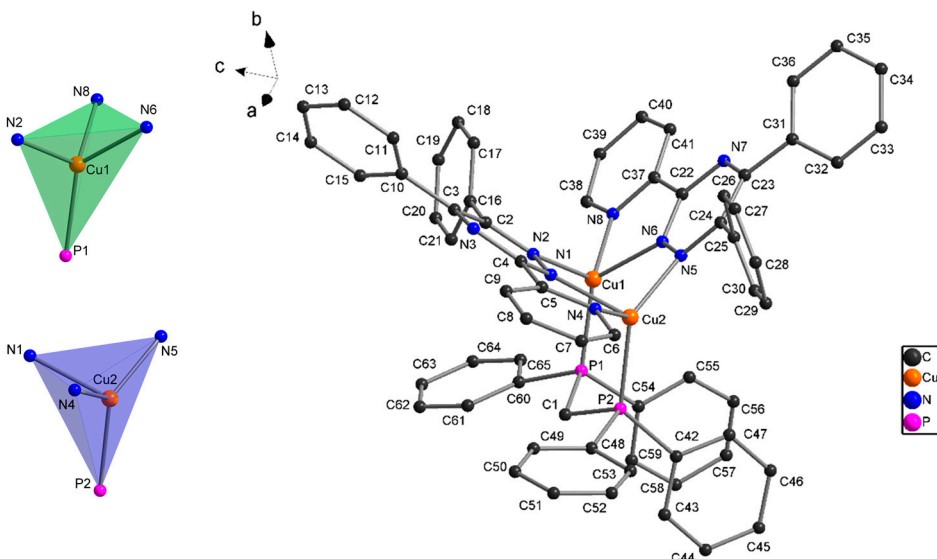

**Figure 1.** Perspective view of the molecular structure and metal coordination polyhedra for $[Cu_2(L)_2dppm]^+$ in the crystal structure of **1**. Hydrogen atoms and counterions were omitted for clarity.

**Table 2.** Experimental (scXRD) and calculated (M06/6-31G(d)+DZVP) data of distances [Å] and angles [°] relevant to the coordination geometries of copper(I) atoms in the complex.

| Distances (Å) and Angles (°) | Experimental | DFT | $\Delta d/\Delta\angle$ |
|:---:|:---:|:---:|:---:|
| Cu(1)-N(8) | 2.047(3) | 2.094 | 0.0470 |
| Cu(1)-N(2) | 2.063(3) | 2.111 | 0.0480 |
| Cu(1)-N(6) | 2.072(3) | 2.169 | 0.0970 |
| Cu(1)-P(1) | 2.183(11) | 2.222 | 0.0389 |
| Cu(2)-N(5) | 2.016(3) | 2.062 | 0.0460 |
| Cu(2)-N(4) | 2.101(3) | 2.132 | 0.0310 |
| Cu(2)-N(1) | 2.136(3) | 2.163 | 0.0270 |
| Cu(2)-P(2) | 2.210(11) | 2.224 | 0.0138 |
| N(8)-Cu(1)-N(2) | 111.85(13) | 112.26 | 0.41 |
| N(8)-Cu(1)-N(6) | 79.00(13) | 77.07 | 1.92 |
| N(2)-Cu(1)-N(6) | 95.50(13) | 94.65 | 0.84 |
| N(8)-Cu(1)-P(1) | 129.33(10) | 130.13 | 0.80 |
| N(2)-Cu(1)-P(1) | 108.25(9) | 112.30 | 4.05 |
| N(6)-Cu(1)-P(1) | 126.45(9) | 119.92 | 6.52 |
| N(5)-Cu(2)-N(4) | 120.54(13) | 117.2 | 3.30 |
| N(5)-Cu(2)-N(1) | 96.31(12) | 95.53 | 0.78 |
| N(4)-Cu(2)-N(1) | 77.67(12) | 76.12 | 1.54 |
| N(5)-Cu(2)-P(2) | 130.45(10) | 129.22 | 1.22 |
| N(4)-Cu(2)-P(2) | 106.72(9) | 111.54 | 4.81 |
| N(1)-Cu(2)-P(2) | 108.05(9) | 109.19 | 1.14 |

The crystallographic study revealed that $[Cu_2(L)_2dppm](PF_6)_2$ (**1**) crystallized in the monoclinic crystal system with space group $P2_1/c$. The asymmetric unit contains two copper(I) atoms as metal centers, two L ligands, one dppm molecule, and two $PF_6^-$ anions. The central Cu(I) ions are embedded in a four-coordinate $CuN_3P$ environment, resulting from coordination by the auxiliary phosphine ligand and triazine L ligands, which adopts the $k^3$-*N,N,N*-tridentate chelate bonding mode of binding via the pyridyl nitrogen and two nitrogen atoms from the triazine group. Thus, two five-membered Cu-N-C-C-N chelate

rings, one six-membered Cu-N-N-Cu-N-N chelate ring, and one seven-membered Cu-N-N-Cu-P-C-P chelate ring are observed in the molecular structure (see Figure S4) with Cu-N and Cu-P bond lengths in the range of 2.0160(3)–2.1360(3) Å and 2.1831(11)–2.2102(11) Å, respectively (Table 2). The bond angles at Cu(I) vary from 77.67(12) to 130.45(10)°, with the smallest value corresponding to the N-Cu-N angle in the five-membered chelate rings formed in the title compound. The angle [N(5)-Cu(2)-P(2)] was the largest due to the sterically demanding dppm ligand chelate ring. A comparison of the bite angle (N-Cu-N) of the triazine ligands with the bond angle calculated (M06/6-31G(d)+DZVP level) agrees well for Cu(1) and Cu(2) ions (see Table 2).

The main distortion of the tetrahedral geometry originates from the small N(1)-Cu(1)-N(4) and N(6)-Cu(2)-N(8) bite angles of the chelating triazine ligands [77.67(12)° and 79.00(13)°, respectively]. The distortion of the tetrahedral geometry around the Cu(I) centers can also be seen from the dihedral angle formed between the two five-membered chelate rings at 74.18° (Figure 1). The coordination geometry is best described as distorted trigonal pyramidal, as indicated by the $\tau_4$-values of 0.74 for Cu(1) and 0.77 for Cu(2) [35]. The geometries are similar to those reported previously for [Cu2(N^N)2(dppm)2](BF4)2 (N^N=2-(2-tert-butyl-tetrazol-5-yl)pyridine) [36], [{Cu(pypzH)}2(μ-dppm)2](ClO4)2 (pypzH=3-(2′-pyridyl)pyrazole) [37] and [Cu(N^N)(PPh3)2]NO3 (N^N = 5,6-diphenyl-3-pyridin-2-yl-[1,2,4]triazine) [38]. In **1**, the intramolecular Cu(1)···Cu(2) distance is 3.217 Å; this value is longer than the sum of van der Waals radii of Cu (2.8 Å), which does not favor metal–metal interaction. In this complex, two face-to-face intramolecular π-stacking interactions stabilize the structure further (Figure S5). The first π-contact is within one dppm ligand (angle between ring planes = 12.4°, centroid···ring plane = 3.62 Å, distance between ring centroids = 3.68 Å). The second is between the phenyl ring of the dppm ligand and the pyridine ring (angle between ring planes = 16.1°, centroid···ring plane = 3.69 Å, distance between ring centroids = 3.80 Å) [39,40].

A detailed analysis of the single-crystal structure of the complex reveals a 3D hydrogen bond array, where two different dimeric units are formed through C-H···π and π···π contacts [39,41] between two [Cu2(L)2dppm]+ cations (Figures S6 and S7). A series of additional C−H···N, C-H···F, C-H···π and π···π interactions between the components of the complex further stabilized the system. The details of these supramolecular interactions are summarized in Table S1. All distances and angles are in agreement with the data found for previously reported structures [29,30,33,42,43].

### 2.3. Hirshfeld Surface Analysis

Hirshfeld surface analysis detects different intermolecular interactions in crystal packing [44,45]. For this purpose, the CIF file was used to generate the Hirshfeld surfaces and fingerprint plots employing the CrystalExplorer program. The red–blue–white color scheme is utilized for quantifying the intermolecular interactions and provides a resource to analyze the zones of strong donor–acceptor interactions [44]. The Hirshfeld surface of the title complex is mapped over the $d_{norm}$ (0.5 to 1.5), curvedness, and shape index (Figure 2). These reports point out the corresponding distances to the closest atom inside the surface (di) and to the closest atom outside the surface (de). The differences reveal alterations in the packing of the structures. Intermolecular π···π interactions between neighboring molecules in the structures of molecular crystals give rise to patches in the curvedness map [46]. The curvedness plots (−4.0 to +4.0) of the complex show only slightly flattened surface patches above either side of the aromatic rings from the L ligand, indicating that the π···π contacts are relatively weak and are significantly face-to-face displaced (Figure 2b). Maps of the shape index are more sensitive to subtle changes in the electron density surrounding the molecules [45,46]. The shape index curve exhibits a complementary red (pit)—blue (bump) color that corresponds to the negative and positive surface property value, respectively, with the former representing the location of an acceptor atom and the latter pointing towards a donor atom and being involved in C−H···N, C-H···F, C-H···π and π···π interactions, in agreement with the observations in the scXRD section (Figure 2c).

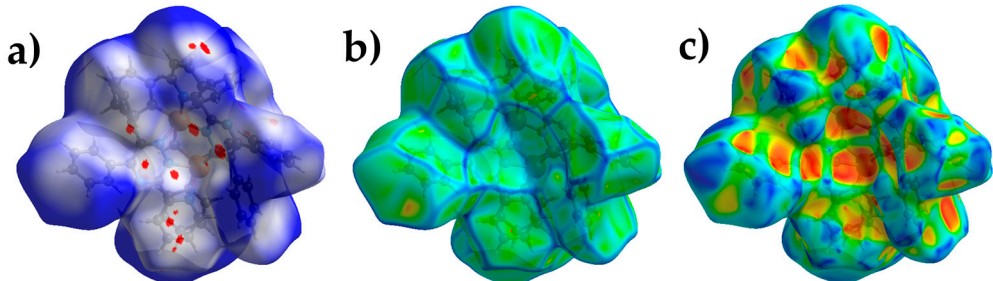

**Figure 2.** Hirshfeld surface for the title complex mapped with $d_{norm}$ (**a**), curvedness (**b**), and shape index (**c**).

The dominant interactions observed in complex **1** are H···H (55.4%), H···C (19%), H···F (18%), and C···F (%), which appear as red spots on the $d_{norm}$ surface in Figure 2a.

Furthermore, the intermolecular interactions in the complex are represented in the 2D plots shown in Figure 3 and Figure S8, respectively. The fingerprints around 1.6–1.8 ($d_i$, $d_e$) vary from a blue tone to a slightly green color associated with the C···C contacts from π···π interactions [47,48]. The H···F/F···H and H···N/N···H interactions appear as distinct spikes in the fingerprint plot and comprise 18% and 1.5%, respectively, of the total Hirshfeld surface for complex **1** (Figure 3 and Figure S8). The more dispersed zones in the blue color correspond mainly to H···H (55.4%) van der Waals contacts. The significant contribution of H···H contacts indicates that aside from the hydrogen bonding interactions, van der Waals contacts are relevant for the molecular packing of the components in the crystal structure.

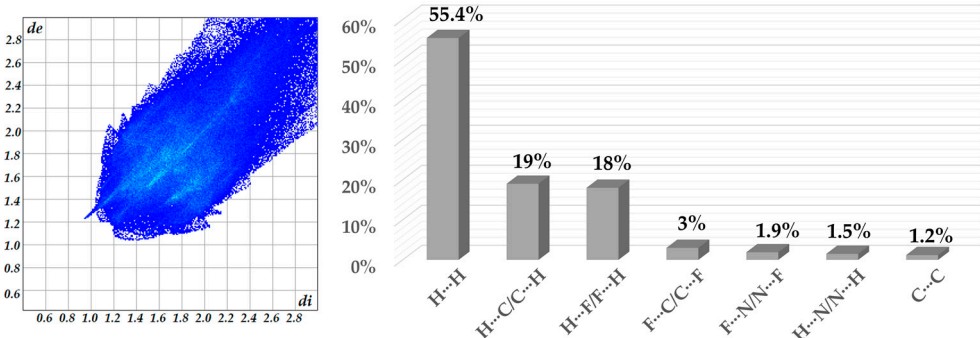

**Figure 3.** 2D full fingerprint plot and percentage contributions of the Hirshfeld surface area for the complex.

### 2.4. Analysis by DFT and UV–Vis

The complex's molecular structures and electronic properties are calculated by DFT [49,50] and TD-DFT methods [51,52]. The calculations were carried out using the M06 hybrid-meta-GGA function in combination with the base sets 6–31G(d) (for C, H, N, and P atoms) and DZVP (Cu atom) with an IEF-PCM in ethanol [53–55]. X-ray crystallographic analysis determined the ground-state geometries from the experimental structure. Notably, the deviations between the simulated molecular structure in solution and the solid-state structure are less than 0.097 Å and 6.52°, respectively (see Table 2).

It is well known that frontier molecular orbital analysis is a potential tool for studying the molecular electronic charge mobility, chemical reactivity, kinetic stability of molecules, and electronic transitions in molecules. The energy difference between the HOMO and LUMO is an important parameter in establishing the photophysical and electrical properties of organic and inorganic materials [29,30,33].

Since the key absorption processes are HOMO→LUMO transitions, it is important to establish the corresponding separate states of charge with the HOMO located in a donor moiety and the LUMO in an acceptor unit. Thus, the isodensity plots of the frontier

molecular orbitals (FMOs) for the asymmetric unit of $[Cu_2(L)_2dppm]^{2+}$ at the M06/6-31G(d) + DZVP level of theory of calculation show charge transfer (HOMO→LUMO) over the entire π-system of the compound and the copper metal center. The energy of the highest occupied molecular orbital ($E_{HOMO}$) is −6.656 eV, and the energy of the lowest unoccupied molecular orbital ($E_{LUMO}$) is −3.034 eV, giving $\Delta E_{(LUMO-HOMO)}$ = 3.623 eV (Figure 4). As shown in Figure 4 and Figure S9, the HOMO orbital is mainly concentrated in the copper metal centers and the two P-atoms, while the LUMO electron density is mainly distributed in the L ligands. Furthermore, Figure S7 shows that HOMO—4, HOMO—5, and HOMO—6 orbitals are distributed over the L and dppm ligands, while LUMO + 1 and LUMO + orbitals are distributed only in the triazine ligands. The HOMO and LUMO energy levels of $[Cu_2(L)_2dppm]^{2+}$ are shown in Figure 4. We found that the energy levels of the compound are appropriate for the DSSC system containing $TiO_2$ because the LUMO levels lay above the conduction band of the $TiO_2$ semiconductor (−4.40 eV). Therefore, there is an efficient electron donation, and the HOMO energy levels lay below that of the $I^-/I_3^-$ redox electrolyte (−4.60 eV), which can be improved (about −0.3 V) by adding 4-tert-butyl pyridine (TBP) to the $I^-/I_3^-$ redox electrolyte, contributing to dye regeneration [56,57].

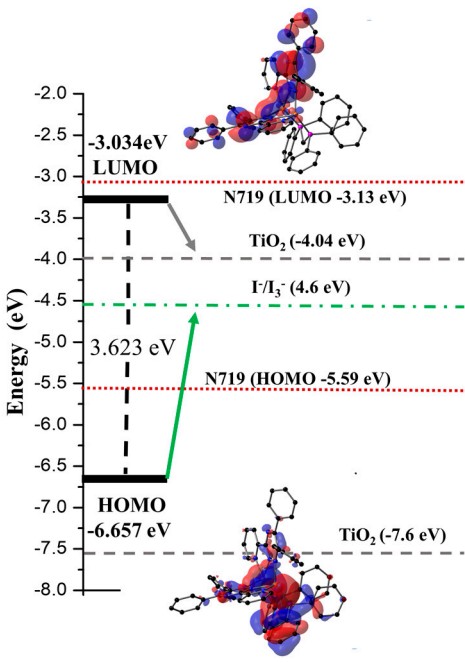

**Figure 4.** A plot of the molecular orbitals involved in the lowest-lying electronic absorption transitions in the complex.

Finally, Figure 4 shows that the LUMO was delocalized in the entire chromophore (L ligands), which might enhance the electron coupling with $TiO_2$ and the electron extraction pathways.

The experimental and calculated UV–Vis absorption spectra of the title compound are shown in Figure 5. The experimental spectrum was measured from a $2.0 \times 10^{-5}$ M solution in EtOH at room temperature. The oscillator strength (f) is a parameter that quantifies the probability of electron transitions and is calculated based on the TD-DFT/M06/6-31G(d) + DZVP level of theory. The results of the TD-DFT calculation indicate three major transitions for complex $[Cu_2(L)_2dppm]^{2+}$ (Figures 5 and S10; Table 3, of which the most intense band at 446 nm (f = 0.0922) is due to the HOMO→LUMO transition having an MLCT/XLCT/LLCT character. This excitation is consistent with the experimental spectrum's broad band centered at 410 nm (ε = 13,150 $M^{-1}$ $cm^{-1}$, see Figure 5 and Table 3).

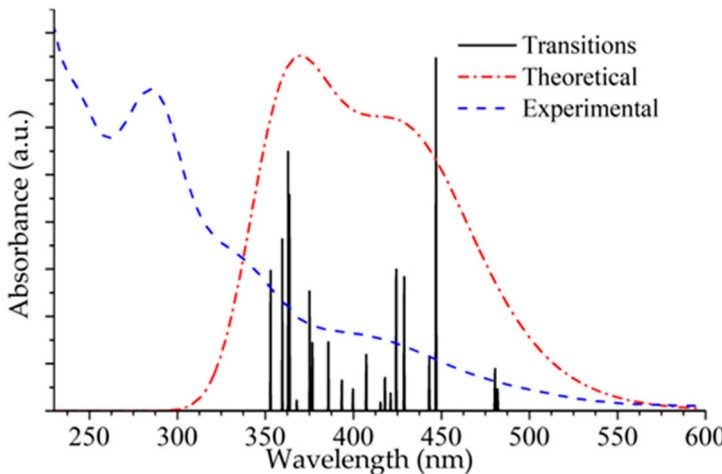

**Figure 5.** Experimental and calculated absorption spectra of $[Cu_2(L)_2dppm](PF_6)_2$.

**Table 3.** Principal electronic excited states calculated by time-dependent density functional theory (TD-DFT) at M06/6-31G(d)+DZVP level of calculation, together with the experimental values *.

| $\lambda_{DFT}$ (nm) E (eV) | $\lambda_{Exp}$ (nm) | Oscillator Strength | Transition (CI Coef.) | Character |
|---|---|---|---|---|
| 446(2.77) | 410 | 0.0922 | HOMO→LUMO (41%)<br>HOMO-2→LUMO (25%) | MLCT/XLCT/LLCT |
| 428(2.89) | | 0.035 | HOMO→LUMO+1 (29%)<br>HOMO-2→LUMO+1 (21%) | MLCT/XLCT/LLCT |
| 424(2.92) | | 0.037 | HOMO-1→LUMO+1 (29%) | MLCT/XLCT/LLCT |
| 363(2.41) | 338 | 0.0565 | HOMO-3→LUMO+3 (34%)<br>HOMO-6→LUMO (21%) | LLCT |
| 375(3.31) | | 0.0313 | HOMO-6→LUMO (26%) | LLCT |
| 359(3.45) | 286 | 0.0449 | HOMO-10→LUMO (20%)<br>HOMO-6→LUMO (18%) | LLCT |
| 352(3.51) | | 0.0367 | HOMO-5→LUMO+1 (36%)<br>HOMO-4→LUMO (21%) | MLCT/XLCT/LLCT |

* Metal-to-ligand charge transfer (MLCT); Ligand-to-ligand charge transfer (LLCT); Phosphine-to-ligand charge transfer (XLCT).

The calculated spectrum displays two additional bands at 363 nm (f = 0.0565) and 359 nm (f = 0.0449), which are assigned to HOMO-3→LUMO+3/HOMO-6→LUMO and HOMO-10→LUMO/HOMO-6→LUMO transitions, respectively. These transitions imply that intramolecular charge transfer takes place [29,30,58]; the band at 363 nm can be related to the broad experimental absorption band found at 338 nm ($\varepsilon$ = 26,700 $M^{-1}$ $cm^{-1}$, Figure 5 and Figure S10) and has an LLCT character. The experimental band centered at 286 nm ($\varepsilon$ = 56,700 $M^{-1}$ $cm^{-1}$) was assigned to $\pi\rightarrow\pi^*$ transitions, having an LLCT character. A detailed assignment of the TD-DFT calculations in terms of FMOs is included in Table 4.

**Table 4.** Summary electrochemical data of the $[Cu_2(L)_2dppm](PF_6)_2$ complex in acetonitrile *.

| | $E_{ox}$ [V] | $E_{red}$ [V] | $E_{HOMO}$ [eV] | $E_{LUMO}$ [eV] | $E_{HOMO/DFT}$ [eV] | $E_{LUMO/DFT}$ [eV] | $\Delta_E$ [eV] | $\Delta_{E/DFT}$ [eV] |
|---|---|---|---|---|---|---|---|---|
| Complex | 1.15 | −1.61 | −5.53 | −2.77 | −6.657 | −3.034 | 2.76 | 3.623 |

* The CV was recorded in acetonitrile solution for the complex (5 × $10^{-3}$ M) with 0.1 M of $NBu_4PF_6$ (T = 298 K, scan rate = 25 mV $s^{-1}$), $\Delta_E$ [eV] = $E_{LUMO}$ − $E_{HOMO}$.

## 2.5. Electrochemical Properties

The electrochemical properties of the dinuclear complex were assessed at 298 K for solutions in acetonitrile by cyclic voltammetry (CV), using 0.1 M of tetrabutylammonium hexafluorophosphate ($NBu_4PF_6$) as a supporting electrolyte. The analyzed data are found in Table 4, and the CV is shown in Figure S11. The complex showed irreversible oxidation and reduction waves. The first oxidation ($E_{pa}$ = 0.97 V) corresponds to the Cu(I)/Cu(II) redox couple with a significant P^P character, indicating stronger structural rigidity [30,59]. The compound shows a second irreversible oxidation wave ($E_{ox}$ = 1.15 V) assigned to the oxidation of the second copper center, revealing the expected electronic communication between the two metals [59]. The oxidation potential (+0.97 V and +1.15 V) is within the range reported for copper(I)-pyridyl complexes [29,30,59,60]. The first reduction event ($E_{red1}$ = −0.81 V) is centered on the pyridine ring of the L ligand; a second reduction wave at −1.96 V is assigned to a second reduction of the L ligand [30,59]. Based on the reduction potentials, the HOMO and LUMO energy levels were calculated using Equation (1) [61,62]:

$$E_{HOMO} \text{ (or } E_{LUMO}) = -4.8 - [(E_{peak\ potential} - E_{1/2} \text{ (ferrocene)}] \tag{1}$$

where $E_{peak\ potential}$ corresponds to the maximum and minimum peak potential and $E_{1/2}$ is the half-wave potential of ferrocene (0.42 V), which was used as a reference. The resulting value for the HOMO orbital (−5.53 eV) agrees with the values obtained by the DFT calculations, with −6.65 eV. Due to the irreversibility of the redox process, it was not possible to obtain a good approach for the LUMO value.

The HOMO and LUMO energy levels in complex **1**, estimated from cyclic voltammetry (CV) measurements, are −5.53 and −2.77 eV, respectively. They are consistent with the DFT results in that the energy levels of the synthesized compound are suitable for electron injection and dye regeneration thermodynamically (see Scheme S1).

## 2.6. TGA Analysis

To investigate the thermal behavior of **1**, thermogravimetric analysis (TGA) was performed over the temperature range of 30–800 °C under an $N_2$ atmosphere for a crystalline sample with a heating rate of 20 °C $min^{-1}$ (see Figure 6). The TGA graph indicates the first weight loss (found, 10.0%; theoretical, 10.2%), in the 300−350 °C range, attributed to the loss of one $PF_6$ ion. The second step in the temperature range of 350 to 450 °C corresponds to the loss of two L ligands, one dppm molecule, and one $PF_6$ ion. The observed weight loss of 80.3% agrees with the calculated value (80.9%). The residual framework starts to decompose beyond 450 °C with a series of complicated weight losses and does not stop until heating ends at 800 °C.

## 2.7. Application in DSSCs

When the crystal structure of the complex is considered, it is found that structural rigidity and molecular packing are critical factors for electrochemical properties because they facilitate electronic transfer [56]. In the complex, the center Cu(I) metal chelates with two N electron donors of the L ligand and forms two five-membered Cu-N-C-C-N chelate rings, which are adjacent to the triazine and benzene rings (see Figure S4). The five-membered chelate ring, triazine ring, and benzene ring are almost coplanar; thereby, the rigidity of the molecular structure is increased. Furthermore, complex **1** extends as supermolecules through C−H⋯N, C-H⋯F, C-H⋯$\pi$ and $\pi$⋯$\pi$ interactions (see Figures S6 and S7, and Hirshfeld surface analysis), which make the electron-donating ability of the title complex better when they are used in DSSCs as a co-sensitizer.

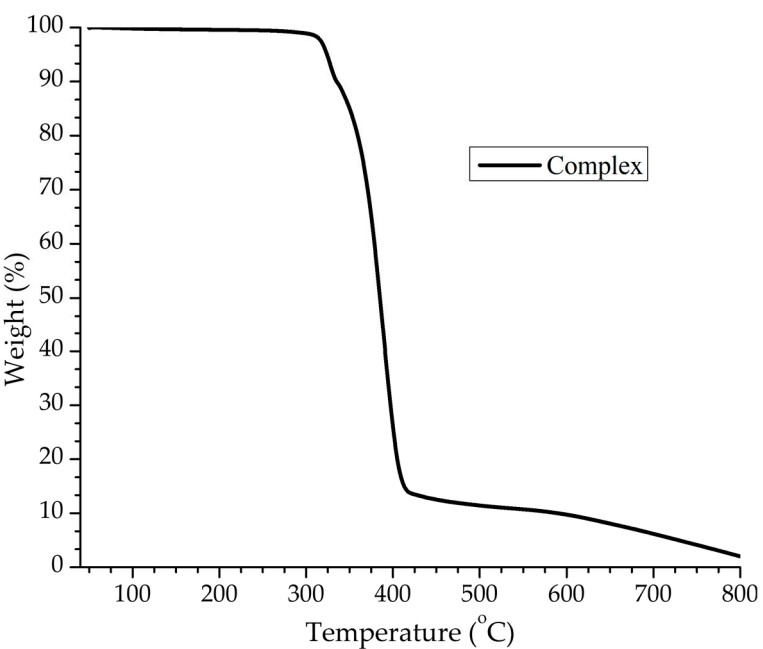

**Figure 6.** TGA curve of complex **1**.

In addition, UV–Vis absorption spectra of ligand L, complex **1**, and N719 were measured in ethanol at room temperature (Figure S10). The complex showed a metal-to-ligand charge transfer (MLCT) absorption band between 350 and 550 nm. Compared to the commercially available N719 ruthenium complex, which absorbs in the 330 to 600 nm range [63], the complex could achieve absorption at a low wavelength when used as a co-sensitizer in DSSCs. A higher molar extinction coefficient at 338 nm ($\varepsilon$ = 26,700 $M^{-1}$ $cm^{-1}$) indicates that complex **1** possesses a higher light-harvesting ability in this wavelength region compared with N719 and $I_3^-$ (25,000 $M^{-1}$ $cm^{-1}$) [56]. Hence, it can be predicted that the photon lost due to the light absorption by $I_3^-$ will be suppressed by using **1** as a co-sensitizer and co-adsorbent due to the competition between **1** and $I_3^-$ to absorb light. To evidence this hypothesis, two DSSC devices were developed; the first was sensitized with N719 alone and was used as a control, and the second was co-sensitized with a 1:1 ($w/w$) mixture of complex **1** and N719. Notably, the amount of N719 used in the co-sensitized device was only half (50%) that used in the control DSSC device.

The current–voltage (J–V) characteristics of the DSSC device based on the N719 and complex/N719 photoanodes are shown in Figure 7, and the efficiencies of the corresponding cells are summarized in Table 5. Under standard global AM1.5 solar irradiation conditions, the electrode performance ($\eta_{rel}$) of the complex/N719/$TiO_2$ co-sensitized solar cell decreased by 7.63%, representing an acceptable value because the amount of N719 was lower. These results suggest that the co-sensitization of $TiO_2$/N719 photoelectrodes with the Cu(I) complex is an option to reduce the amount of N719 dye, cutting costs while having a minor impact on the efficiency of DSSCs.

**Table 5.** J–V performance of DSSCs based on different photoelectrodes.

| Dyes | $Jsc$ (mA/cm$^2$) | $Voc$ (V) | $FF$ (%) | $\eta$ (%) | $\eta_{Relative}$ (%) |
|---|---|---|---|---|---|
| [a] Complex/N719 (1:1) | 5.095 | 0.757 | 52.7 | 2.03 | 92.27 |
| N719 | 6.030 | 0.770 | 47.3 | 2.2 | 100 |

*Jsc* = short circuit current, $V_{OC}$ = open circuit potential, *FF* = fill factor, $\eta$ = power conversion efficiency. [a] The electrode based on the dye combination 1/N719 contains only 50% of N719 compared to the control experiment with only N719.

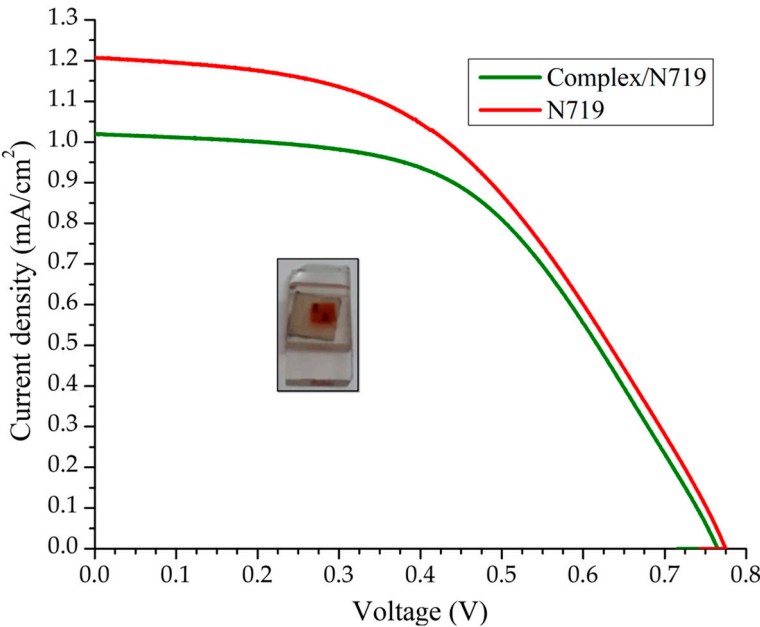

**Figure 7.** J–V curves for DSSCs of N719 and complex/N719. Inset: Photo of the DSSC device containing dye complex/N719.

## 3. Conclusions

A new dinuclear copper(I) compound based on 3-(2-Pyridyl)-5,6-diphenyl-1,2,4-triazine as a chelating ligand and auxiliary phosphine Bis(diphenylphosphino)methane was prepared and characterized using a variety of analytic techniques. Additionally, computational studies were used to understand the electronic characteristics of the compound. The structural data revealed that the complex possesses a distorted trigonal pyramidal geometry and a variety of supramolecular interactions, such as C-H⋯N, C-H⋯π and π⋯π, which stabilize the crystalline structure. A comparison of experimental (SCXRD analysis) and calculated (DFT/M06/6-31G(d)+DZVP) bond lengths and bond angles showed excellent agreement, with variations less than 0.097 Å and 6.52°, respectively. Complexes **1** displays a low-intensity band at 410 nm, corresponding to MLCT transitions, consistent with the theoretical calculation realized with EtOH. According to the voltammetry analysis, the complex shows irreversible oxidation processes, which constitutes a drawback for the regeneration of dyes within DSSC devices. Devices based on $TiO_2$/N719 and co-sensitized with the complex produce overall efficiencies of 92.27%, which is slightly lower than the reference device but employs only half the amount of the expensive and more toxic ruthenium dye (N719). The results of the current report shed some light on the design of co-sensitizers for the fabrication of new DSSCs with low prices and higher access to the Earth-abundant copper-based precursors.

Finally, the current findings suggest moving in two directions: (i) replacing the dppm ligand for a ligand that has anchoring functional groups, to seek an optimal balance between the efficiency of electron injection and the stability of the dye or (ii) replacing the diphenylphosphine ligand with a more suitable ligand to generate a beneficial impact on the absorption properties of the sensitizer.

**Supplementary Materials:** The following supporting information can be downloaded at https://www.mdpi.com/article/10.3390/inorganics11100379/s1: Materials and Methods, Figures S1–S3, IR, [1]H-NMR and [31]P-NMR spectra of **1**; Table S1, geometries of intermolecular hydrogen bonds and π⋯π contacts in complex **1**. Figure S4, perspective views of $[Cu_2(L)_2dppm]^+$ in the crystal structure of 1, showing (a) two five-membered Cu-N-C-C-N, (b) one six-membered Cu-N-N-Cu-N-N and (c) one seven-membered Cu-N-N-Cu-P-C-P chelate rings; Figure S5, intramolecular π⋯π interactions in the crystal structure of **1**; Figure S6, intermolecular C-H⋯π and π-stacking interactions between pair of $[Cu_2(L)_2dppm]^+$ cations; Figure S7, perspective view of the three-dimensional

(3D) hydrogen-bonded network in the crystal structure of complex **1**, formed through C-H···N, C-H···F, C-H···$\pi$ and $\pi$···$\pi$ interactions; Figure S8, percentages of intermolecular interactions in the fingerprint plot for complex **1**; Figure S9, HOMO and LUMO frontier orbital plots of the title complex on TD-DFT calculations; Figure S10, UV–Vis absorption spectra of complex, free ligand L and N719 recorded in $2 \times 10^{-5}$ mol/L solution in ethanol; Figure S11, cyclic voltammogram of $[Cu_2(L)_2dppm](PF_6)_2$ ($5 \times 10^{-3}$ M) in acetonitrile at $T$ = 298 K using $NBu_4PF_6$ (0.1 M) as supporting electrolyte (scan rate = 20 mVs$^{-1}$). Scheme S1. Schematic illustration of the operational principle of DSSC and energy diagram of HOMO and LUMO levels for dyes compared to the energy levels for $TiO_2$. References [64–79] are cited in the supplementary materials.

**Author Contributions:** C.A.P., investigation, methodology, material analysis, and writing—original draft; J.J.C.-G., conceptualization, visualization, project administration, writing—original draft, and writing—review, and editing; E.A.R.-S., investigation, and methodology; A.C.-E., conceptualization, formal analysis, material analysis, and writing—original draft; R.S.-R., conceptualization, software and writing—original draft; J.B.-L., conceptualization and software; J.J.G., X-ray analysis, visualization, writing—original draft, writing—review and editing; M.F.-Á., X-ray and material analysis; V.M.-S., investigation, methodology; D.G.-M., software and validation. All authors have read and agreed to the published version of the manuscript.

**Funding:** This work received financial support from Universidad Autónoma de Sinaloa, México (DGIP-PRO-A2-023) and from Consejo Nacional de Ciencia y Tecnología (CONACyT) in The form of a scholarship granted to C.A.P. (701513).

**Institutional Review Board Statement:** Not applicable.

**Informed Consent Statement:** Not applicable.

**Data Availability Statement:** Not applicable.

**Acknowledgments:** The authors gratefully acknowledge access to the instrumental support in the USAII Facultad de Química. UNAM, México.

**Conflicts of Interest:** The authors declare no conflict of interest.

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
