# Peer review of "Synthesis of a New Dinuclear Cu(I) Complex with a Triazine Ligand and Diphenylphosphine Methane: X-ray Structure, Optical Properties, DFT Calculations, and Application in DSSCs"

_inorganics, doi:10.3390/inorganics11100379_

Round 1

Reviewer 1 Report

In this manuscript, J. Campos-Gaxiola and co-workers describe the use of a di-nuclear Cu(I) triazine diphenylphosphine complex as a co-sensitizer (in combination with N719) in a dye-sensitized solar cell.

The new complex is deeply characterized and the methods are well described. The authors give good explanations of all the experimental behaviours and the proposed references are appropriate.

Overall the work is interesting and clear,  I found only two points that are not clear to me and I think that merit further explanation in the text:

1) usually the presence of carboxylic  or phosphonic  acid groups on the dye is necessary to anchor it to the TiO2 surface. The reported complexes do not present such anchoring group. How can the dye bind to the titania surface?

2) In the preparation of the co-sensitized electrode, how can the authors be sure about 1:1 ratio  complex:N719, since the electrode is rinsed and the copper complex does not have anchoring groups?

In table 1 is reported for the ligand n(C-H) = 3055 and for dppm n(C-H) = 3050 while in the supporting information (Figure S1) is the opposite ( for ligand 3050)

In my opinion the work is of interest for the people that work in this field  and merits publication in Inorganics

Reviewer 2 Report

Dear authors, I have read your manuscript on a new dinucler copper dye for DSSCs. The synthesis is fine and the complex is well characterized.

In my opinion there are two problems that lead to a poor efficiency of the cell: the absence of anchoring groups and the irreversible oxidation process. 

-Could you please comment together the UV-vis spectra of your dye and N719, is there a better coverage of the solar spectrum using both dyes? 

-How can you say that the % of N719 anchored to TiO2 is 50% when you use also your dye? When you wash the cell, how can you say that your complex is still in the cell?

-In the conclusion I suggest to underline the future improvement (such as anchoring groups) to enhance the performence of the DSSC.

In conclusion, I think that the manuscript can be accepted after minor revisions, reported above.

Reviewer 3 Report

The paper is concerned with Synthesis of a New Dinuclear Cu(I) Complex with Triazine Ligand and diphenylphosphine methane: X-ray Structure, Optical properties, DFT calculations and application in DSSC. The paper covers some important issues but some points should be corrected. I recommend the manuscript for publication after major revision.

1. Page 1: Abstract: Need to revise properly. Add results, conclusion, and prospects of this review. Confirm the word limit for the abstract.

2. In the introduction, it should be discussed detailed about the difference of this material with other materials.

3. The resolution of images must be improved during the revision stage.

4. Need to add the exact mechanisms for each sensing application

5. Abbreviations must be defined at their first mention (abstract and text separately).

6. Some DFT refs could be cited, such as Theor Chem Acc, 142 (2023) 78; Theor. Chem. Acc. 2022, 141, 68 and , Monatsh. Chem, 2019, 150, 1355–1364.

7. The packing mode of such complexes could be discussed in detail.

8. Pls provide the IR, TGA and PXRD for all complex.

check

Round 2

Reviewer 3 Report

accept